## Comment

behaviour/neuroscience/psychology

**Author for correspondence:**
Jonathan W. Peirce
e-mail: jonathan.peirce@nottingham.ac.uk

# Do sheep really recognize faces? Evidence from previous studies. Response to 'Sheep recognize familiar and unfamiliar human faces from two-dimensional images'

## Jonathan W. Peirce

School of Psychology, University of Nottingham, University Park, Nottingham NG7 2RD, UK

JWP, 0000-0002-9504-4342

Knolle *et al.* [1] make a number of claims about the processing of sheep faces, some of which have been questioned by Towler *et al.* [2]. The question is whether Knolle *et al.* claimed more about the abilities of the sheep with respect to face recognition than was warranted from their data. Their points are fair in as much as the data provided by Knolle *et al.* do not support all of the claims, but there are further published data, from studies in Keith Kendrick's lab in the late 1990s that do support the conclusions rather more strongly. Kendrick's lab tested sheep on more faces, in a wider range of experimental conditions, both behaviourally and using single-unit recordings in temporal cortex [3–7], and showed a number of ways in which face processing in sheep is really rather similar to that in humans.

Towler *et al.* focus particularly on the claim that '*sheep have advanced face-recognition abilities, comparable with those of humans*' and questions whether this is true either in terms of comparable performance or comparable patterns of behaviour. To the former possibility, the answer is surely that the sheep do *not* operate at the same performance level and Towler *et al.* provide various sources of data to make that clear. This is surely the less interesting question, however. The fact that overall performance is lower in sheep could have many explanations that are relatively trivial to a psychologist or neuroscientist. To name just a few, if the sheep attention span, memory capacity, visual acuity or any combination thereof were quantitatively poorer than those of a human then we would expect their overall performance to be poorer on a task, but this finding seems uninteresting. Presumably, however, Knolle *et al.* were not intending to claim that sheep have the same level of performance as humans. As it

happens, in the studies from Keith Kendrick's lab, the sheep were challenged further and were found, for instance, to remember the visual identities of 50 individuals over a period of roughly two years [3]. Still, this does not suggest that their absolute performance is anywhere near the same level as their human counterparts.

The second question seems rather more interesting: Do sheep show similar *patterns* of performance, indicative (but only indicative) of similar mechanisms? Towler *et al.* suggest that Knolle *et al.*'s evidence is also weak on this point. This, however, is where Kendrick's lab has contributed a number of findings that shed greater light on the issue.

For instance, the sheep of Knolle *et al.* performed as well on previously unknown sheep faces as they did on highly familiar human faces (the animal handler). Towler point out that this is quite dissimilar to humans, for whom there is a substantial advantage in recognizing more familiar individuals. There are two issues here: the first is that Peirce *et al.* [4] show a better comparison than the Knolle *et al.* paper: we tested sheep with faces from their *own flock* ('familiar') and from *another flock* ('unfamiliar'). We showed not only the expected difference in performance (better at discriminations with previously familiar faces) but also in the pattern of that perception (a greater use of internal and configural cues, and a left-visual-field advantage, when processing faces of their own flock; [4]). Peirce *et al.* [5] went on to show a further expertise effect, akin to the other-race effect, that human faces are much harder for sheep to discriminate. While sheep were above chance at discriminating human faces from photographs, discriminations of even familiar humans were harder for them to learn than sheep face discriminations, and did not show the same 'expert' hallmarks of inversion effects, configural coding or visual field bias [5]. The suggestion that the sheep in the Knolle *et al.* study were poorer with highly familiar individuals is confounded by the fact that those familiar individuals are of a different species. Whether the 'race effect' should outweigh the 'familiarity effect' is likely dependent on many conditions, and in their study the effects appeared roughly to balance, but the two effects have already been measured independently (inter-species effect in Peirce *et al.* [5] and the familiarity effects in Peirce *et al.* [4]) and show a similar pattern to the effects in humans.

Interestingly, Towler *et al.* and Knolle *et al.* both assert that, by training on full-frontal faces and testing on profile views, Knolle *et al.*'s work has shown the most convincing case for genuine face recognition outside humans. This is incorrect on two scores. Firstly, they were not the first to show the recognition of a profile after training only on the frontal view; the same finding was presented, albeit briefly, over 15 years earlier [3]. Transfer of learning from frontal to profile view was shown again a few years later by another group [8]. That study also showed a transfer across different 'model' ages as well; having learned to discriminate three-month-old lamb faces, the sheep were above chance at discriminating the same lambs from their one-month snapshot, with no further training.

More importantly, although many authors consider it the key evidence of face processing, merely rotating a face from frontal to profile view does not actually provide very good evidence of genuine face recognition (where the individual understands that the two-dimensional photograph corresponds to a real individual and could identify that individual). A fairly simple computer algorithm could detect sufficient image similarities (skin tone or colour, for instance) to be above chance on the rotated face, and we would not consider those dumb algorithms to possess face recognition.

On the other hand, the familiarity effect shown by Peirce *et al.* [4] indicates that sheep perform better (and use configural cues more) if they had prior *social experience* with the individual in the photograph. The way they discriminated these two-dimensional photographic stimuli depended on their prior experience of the individual in the three-dimensional world. That is very hard to explain by any image-matching algorithm which, almost by definition, cannot have had either social or three-dimensional experience. For prior social experience to impact the performance, in a photographic discrimination task, we have to evoke some notion of continuity of perception across the two domains; the sheep must, on some level, detect this to be the same individual that they have experienced socially. That remains, to my knowledge, the best current evidence for genuine face recognition by a non-human animal.

Towler *et al.* also raise concerns about comparing sheep with humans without including a direct comparison group on the same task. The core aspect of the task is actually a common one in human perception tasks—our sheep were presented with a two-alternative forced-choice task, where their aim is to detect face A in the presence of face B. The stimulus manipulations were similar to those used in human tasks—we generated various stimuli such as 'half-mirrored' faces, 'chimeric' faces (half of one individual with half of another) and inverted faces, and tested whether the faces could still be discriminated following these manipulations. All the findings that we showed had already been found in similar tasks in humans. Since we were never intending to claim that the effects were similar

*in percentages* to the human effects, it isn't clear what would be gained by running humans in the same exact study, and the humans would probably have been at ceiling level performance, rendering the measurements uninformative.

The extent to which the face recognition is 'comparable' comes from the fact that the sheep show effects of face inversion, left hemifield bias, configural coding effects, social familiarity effects and species expertise effects. Whether any of these was the same magnitude, and whether sheep have the same ultimate skill level as humans, was not something that we had considered interesting, and seems very unlikely.

Knolle *et al.* also point out, quite correctly, that we don't know whether this similar pattern of results comes from similar neural mechanisms. The neural substrates of effects, like configural coding in face recognition, are not known and the comparisons being made are purely on similar behavioural patterns.

Nonetheless, it seems reasonable of Knolle *et al.* to claim that sheep face perception is quite advanced and rather similar, in many ways, to face perception in humans, assuming they weren't claiming an equivalent performance. While Towler *et al.* are right to claim that Knolle *et al.*'s own data don't show that in very much detail, it had, as it happens, been shown previously.

Data accessibility. This article has no additional data.

Competing interests. I declare I have no competing interests.

Funding. I received no funding for this study.

Acknowledgements. Many thanks to Michael Hinton, Andrea Leigh and Keith Kendrick, for the many discussions we had about this topic in years past.

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
