## [Reviewer comments · Royal Society Open Science]

Review History

RSOS-182157.R0 (Original submission)

Review form: Reviewer 1 (Sebastian McBride)

Is the manuscript scientifically sound in its present form?

Yes

Are the interpretations and conclusions justified by the results?

Yes

Is the language acceptable?

Yes

Is it clear how to access all supporting data?

Not Applicable

Do you have any ethical concerns with this paper?

No

Have you any concerns about statistical analyses in this paper?

No

Recommendation?

Accept as is

Comments to the Author(s)

The response by Pierce accurately contextualises the study by Knoll et al in relation to previous work carried out in this area. His comments also critically interprets the validity of the Towler et al response and by doing so creates a purposeful and highly useful discussion about what we can and can't yet know about ovine cognition given the data generated from studies generated so far.

Review form: Reviewer 2

Is the manuscript scientifically sound in its present form?

Yes

Are the interpretations and conclusions justified by the results?

Yes

Is the language acceptable?

Yes

Is it clear how to access all supporting data?

Not Applicable

Do you have any ethical concerns with this paper?

No

Have you any concerns about statistical analyses in this paper?

No

Recommendation?

Accept with minor revision (please list in comments)

Comments to the Author(s)

The author claims that his studies in collaboration with K. Kendrick had shown previously (1995-2001) that face recognition in sheep is quite similar to that in humans, as Knolle et al. reported 15 years later.

I agree with the author that Kendrick's lab was a pioneer and did an remarkable job in the study of face recognition in sheep. However, as Towler et al. wrote in their comments, making claims about the similarity of the mechanisms involved in face recognition needs to perform study using an equivalent task between humans and sheep. This should be mentioned and Kendrick's studies did not used such a methodology. In addition, when one comment about the similarity of mechanisms, what mechanism are we talking about? For instance, nothing is known about the neurobiological mechanisms involved in face recognition. It could be possible that cognitive performances are similar between sheep and humans but that face recognition involves similar or different neuronal network is unknown. There is only one study from Kendrick's lab showing that cells of the temporal cortex code for individual recognition (1991).

Please also clarify

- The last sentence on the first paragraph page 2: could you detailed and give references?
- The issue raised on page 2 second paragraph: on which basis the author claims that an image-matching algorithm cannot explained the familiarity effect?
- The study showing that sheep recognize faces of unfamiliar animals at different ages and in different orientations should be cited in the comment (Transfer between views of conspecific faces at different ages or in different orientations by sheep; Ferreira et al., 2004, BEHAV.PROCESS,67, 491-499.)

Decision letter (RSOS-182157.R0)

14-May-2019

Dear Dr Peirce

On behalf of the Editors, I am pleased to inform you that your Manuscript RSOS-182157 entitled "Do sheep really recognize faces? Evidence from previous studies. Response to Knolle et al and Towler et al" has been accepted for publication in Royal Society Open Science subject to minor revision in accordance with the referee suggestions. Please find the referees' comments at the end of this email.

The reviewers and handling editors have recommended publication, but also suggest some minor revisions to your manuscript. Therefore, I invite you to respond to the comments and revise your manuscript.

- Ethics statement

- Data accessibility

If you wish to submit your supporting data or code to Dryad (<http://datadryad.org/>), or modify your current submission to dryad, please use the following link:
<http://datadryad.org/submit?journalID=RSOS&manu=RSOS-182157>

- Competing interests

- Authors' contributions

- Acknowledgements

- Funding statement

Because the schedule for publication is very tight, it is a condition of publication that you submit the revised version of your manuscript before 23-May-2019. Please note that the revision deadline will expire at 00.00am on this date. If you do not think you will be able to meet this date please let me know immediately.

on behalf of Dr Rosalind Arden (Associate Editor) and Kevin Padian (Subject Editor)
openscience@royalsociety.org

Associate Editor Comments to Author (Dr Rosalind Arden):

Associate Editor: 1

Comments to the Author:

Thank you for this contribution to a discussion that has generated some interest. One reviewer has suggested a couple of minor revisions. I would be very grateful if you would make the small adjustment necessary to take into account the suggestions provided, to strengthen the comment. I hope that this will not be onerous and very much look forward to a slightly revised submission.

Reviewer comments to Author:

Reviewer: 1

Comments to the Author(s)

The response by Pierce accurately contextualises the study by Knoll et al in relation to previous work carried out in this area. His comments also critically interprets the validity of the Towler et al response and by doing so creates a purposeful and highly useful discussion about what we can and can't yet know about ovine cognition given the data generated from studies generated so far.

Reviewer: 2

Comments to the Author(s)

The author claims that his studies in collaboration with K. Kendrick had shown previously (1995-2001) that face recognition in sheep is quite similar to that in humans, as Knolle et al. reported 15 years later.

I agree with the author that Kendrick's lab was a pioneer and did an remarkable job in the study of face recognition in sheep. However, as Towler et al. wrote in their comments, making claims about the similarity of the mechanisms involved in face recognition needs to perform study using an equivalent task between humans and sheep. This should be mentioned and Kendrick's studies did not used such a methodology. In addition, when one comment about the similarity of mechanisms, what mechanism are we talking about? For instance, nothing is known about the neurobiological mechanisms involved in face recognition. It could be possible that cognitive performances are similar between sheep and humans but that face recognition involves similar or different neuronal network is unknown. There is only one study from Kendrick's lab showing that cells of the temporal cortex code for individual recognition (1991).

Please also clarify

- The last sentence on the first paragraph page 2: could you detailed and give references?
- The issue raised on page 2 second paragraph: on which basis the author claims that an image-matching algorithm cannot explained the familiarity effect?
- The study showing that sheep recognize faces of unfamiliar animals at different ages and in different orientations should be cited in the comment (Transfer between views of conspecific faces at different ages or in different orientations by sheep; Ferreira et al., 2004, BEHAV.PROCESS,67, 491-499.)

Author's Response to Decision Letter for (RSOS-182157.R0)

See Appendix A.

Decision letter (RSOS-182157.R1)

04-Jun-2019

Dear Dr Peirce,

I am pleased to inform you that your manuscript entitled "Do sheep really recognize faces? Evidence from previous studies. Response to Knolle et al and Towler et al" is now accepted for publication in Royal Society Open Science.

on behalf of Dr Rosalind Arden (Associate Editor) and Kevin Padian (Subject Editor)
openscience@royalsociety.org

Associate Editor Comments to Author (Dr Rosalind Arden):

Thank you for revising the Commentary. It's clearly a topic that engages readers and experts. This will make a very nice contribution to the discussion of sheep cognitive abilities with a focus on face recognition. It was especially good of you to write and revise this Commentary, since your suggestions first arose in the shape of a Review! Thank you for your work.

Appendix A

Associate Editor Comments to Author (Dr Rosalind Arden):

Comments to the Author:

Thank you for this contribution to a discussion that has generated some interest. One reviewer has suggested a couple of minor revisions. I would be very grateful if you would make the small adjustment necessary to take into account the suggestions provided, to strengthen the comment. I hope that this will not be onerous and very much look forward to a slightly revised submission.

Thank you. I have worked to improve the manuscript a little further according to the reviewers' suggestions and in light of the full text of the Towler et al paper, now published. At the point when I wrote the previous draft I had seen the first version of Towler et al but not their final version, which was adapted according to reviewers' comments (including my own). As a result my text was, in places referring to comments they no longer make, and did not include reference to comments that warranted reply.

Reviewer comments to Author:

Reviewer: 1

Comments to the Author(s)

The response by Pierce accurately contextualises the study by Knoll et al in relation to previous work carried out in this area. His comments also critically interprets the validity of the Towler et al response and by doing so creates a purposeful and highly useful discussion about what we can and can't yet know about ovine cognition given the data generated from studies generated so far.

Reviewer: 2

Comments to the Author(s)

The author claims that his studies in collaboration with K. Kendrick had shown previously (1995-2001) that face recognition in sheep is quite similar to that in humans, as Knolle et al. reported 15 years later. I agree with the author that Kendrick's' lab was a pioneer and did an remarkable job in the study of face recognition in sheep. However, as Towler et al. wrote in their comments, making claims about the similarity of the mechanisms involved in face recognition needs to perform study using an equivalent task between humans and sheep. This should be mentioned and Kendrick's studies did not used such a methodology.

I have added text explaining this towards the end of this draft. The basic task we used (2-alternative-forced-choice) and the stimulus manipulations, were very similar to many previous human experiments.

That said, we (or at least I, maybe this isn't true of my colleagues) never meant to claim that sheep match human performance, nor even that the magnitude of, say, a face inversion effect would be the same. I was interested merely in showing "do they show an inversion effect"? Running participants on the same task was therefore not needed and it probably would not have been informative if we had – the humans might well have been at ceiling performance and then not shown an inversion effect on this task, for example. As I had said before, there are many (entirely uninteresting) reasons for differences in performance/magnitude of effects.

In addition, when one comment about the similarity of mechanisms, what mechanism are we talking about? For instance, nothing is known about the neurobiological mechanisms involved in face recognition. It could be possible that cognitive performances are similar between sheep and humans but that face recognition involves similar or different neuronal network is

unknown. There is only one study from Kendrick's lab showing that cells of the temporal cortex code for individual recognition (1991).

Agreed. This is also added explicitly into the discussion.

Please also clarify

- The last sentence on the first paragraph page 2: could you detailed and give references?

In the statement, "the two effects have already been measured independently"? I was meaning to summarise the text above but that clearly wasn't obvious. I have reinserted those references so as to be explicit.

- The issue raised on page 2 second paragraph: on which basis the author claims that an image-matching algorithm cannot explained the familiarity effect?

I hope I have better explained this. The social (3D) experience has altered the 2D perception. The image matching algorithm is not capable of transferring across such different arenas (even if it were capable of handling the 3D input which most image-matching algorithms are not). Indeed, if our candidate image-match algorithm were able to transfer recognition from a live encounter in the field to a flat projection in a lab, then we would start to attribute genuine face recognition.

- The study showing that sheep recognize faces of unfamiliar animals at different ages and in different orientations should be cited in the comment (Transfer between views of conspecific faces at different ages or in different orientations by sheep; Ferreira et al., 2004, BEHAV.PROCESS, 67, 491-499.)

That is an excellent point (I was sure there were other examples of transfer but had forgotten this paper). It is included.